# Aspirin-Triggered Resolvin D1 Reduces Chronic Dust-Induced Lung Pathology without Altering Susceptibility to Dust-Enhanced Carcinogenesis

**DOI:** 10.3390/cancers14081900

**Published:** 2022-04-09

**Authors:** Edward C. Dominguez, Rattapol Phandthong, Matthew Nguyen, Arzu Ulu, Stephanie Guardado, Stefanie Sveiven, Prue Talbot, Tara M. Nordgren

**Affiliations:** 1Environmental Toxicology Graduate Program, University of California Riverside, Riverside, CA 92521, USA; edomi007@ucr.edu (E.C.D.); prue.talbot@ucr.edu (P.T.); 2Division of Biomedical Sciences, School of Medicine, University of California Riverside, Riverside, CA 92521, USA; mnguy171@ucr.edu (M.N.); arzu.ulu@medsch.ucr.edu (A.U.); eguar003@ucr.edu (S.G.); ssvei001@ucr.edu (S.S.); 3Department of Molecular, Cell and Systems Biology, University of California Riverside, Riverside, CA 92521, USA; rattapol.phandthong@ucr.edu; 4Department of Environmental and Radiological Health Sciences, Colorado State University, Fort Collins, CO 80523, USA

**Keywords:** lung inflammation, lung cancer, omega-3 fatty acids, specialized pro-resolving mediators (SPM), organic dust, epithelial to mesenchymal transition (EMT), fibrosis, therapeutic

## Abstract

**Simple Summary:**

Farm workers are at an increased risk of developing acute and chronic lung inflammatory diseases from their everyday exposure to organic dust. Previous investigations have examined the inflammatory effects in mice from single and repetitive exposure to dust from swine confinement facilities, however, no study has explored these effects in a chronic model. To address this research gap, we established a chronic dust exposure mouse model of lung tumorigenesis that was also used to measure the efficacy of omega-3 fatty acid-derived lipid mediators as therapeutics for mitigating these induced responses. Our results from these investigations are the first to evaluate the chronic inflammatory, and carcinogenic effects of these dusts, as well as identify a potential therapeutic strategy for mitigating the inflammatory effects by using an omega-3 fatty acid-derived bioactive lipid mediator.

**Abstract:**

Lung cancer is the leading cause of cancer-related deaths worldwide, with increased risk being associated with unresolved or chronic inflammation. Agricultural and livestock workers endure significant exposure to agricultural dusts on a routine basis; however, the chronic inflammatory and carcinogenic effects of these dust exposure is unclear. We have developed a chronic dust exposure model of lung carcinogenesis in which mice were intranasally challenged three times a week for 24 weeks, using an aqueous dust extract (HDE) made from dust collected in swine confinement facilities. We also treated mice with the omega-3-fatty acid lipid mediator, aspirin-triggered resolvin D1 (AT-RvD1) to provide a novel therapeutic strategy for mitigating the inflammatory and carcinogenic effects of HDE. Exposure to HDE resulted in significant immune cell influx into the lungs, enhanced lung tumorigenesis, severe tissue pathogenesis, and a pro-inflammatory and carcinogenic gene signature, relative to saline-exposed mice. AT-RvD1 treatment mitigated the dust-induced inflammatory response but did not protect against HDE + NNK-enhanced tumorigenesis. Our data suggest that chronic HDE exposure induces a significant inflammatory and pro-carcinogenic response, whereas treatment with AT-RvD1 dampens the inflammatory responses, providing a strong argument for the therapeutic use of AT-RvD1 to mitigate chronic inflammation.

## 1. Introduction

Chronic, unresolved inflammation can lead to an increased risk for the development of chronic lung diseases including fibrosis, COPD, and cancer [1,2]. Exposure to organic dusts, including those from animal confinement facilities, have been shown to elicit potent lung inflammatory responses in exposed individuals, which are attributed to proteases, fungal and bacterial components, such as lipopolysaccharides (LPS) or endotoxin, that exist within these dusts [3,4,5]. The current literature reports conflicting epidemiological studies examining the association between LPS and cancer, with some studies citing an enhancement effect on tumorigenesis, whereas others report a protective effect against tumorigenesis [6,7,8,9]. Acute and repetitive exposure to organic dusts, specifically extracts from swine confinement facilities (HDE), has been shown to induce a potent inflammatory response that includes increased immune cell influx in the lung, secretion of pro-inflammatory cytokines, and inflammatory lung tissue pathology [10,11]. Despite these findings, the chronic effects of these dusts, including the carcinogenic properties, have yet to be fully investigated. Agricultural and farm workers endure exposure to organic dusts on an everyday basis, and as such, they face an increased risk of developing chronic lung disease [12,13,14,15]. Current preventative measures including the use of masks and respirators are available for these occupational groups; however, the use of this personal protective equipment by these communities is not common [16,17,18]. Moreover, there is a need for innovative and effective therapeutic strategies to mitigate or inhibit dust-induced inflammation.

The resolution of inflammation is a key step in the immunoregulatory process to ensure our bodies avoid unresolved inflammation following acute or chronic injury [19,20]. One such way our bodies support this mechanism is through the metabolism of polyunsaturated fatty acids (PUFAs) including the omega-6 PUFA arachidonic acid (AA), and omega-3 PUFAs docosahexaenoic acid (DHA) and eicosapentaenoic acid (EPA). During the initial inflammatory response, AA is enzymatically activated by cyclooxygenase (COX-1 and COX-2) and 5-lipoxygenase (5-LOX) into the pro-inflammatory eicosanoids, prostaglandins, thromboxanes, and leukotrienes [21,22]. During inflammation resolution, there is a lipid mediator class switching mechanism and AA becomes activated by 12-LOX and begins to produce lipoxins, which are a group of specialized pro-resolving mediators (SPMs) that function as biological anti-inflammatory substrates to resolve inflammation cascades and promote a return to tissue homeostasis [23,24]. In addition to lipoxin production, EPA and DHA undergo enzymatic activation by LOX, COX, and cytochrome P450 into SPMs including the maresins, protectins, and resolvins [25,26,27]. SPMs promote the resolution of inflammation by reducing neutrophil influx, increase macrophage recruitment and efferocytosis, and repairing tissue damage at the site of inflammation [19,26,28].

Resolvin D1 and its synthetic epimer, aspirin-triggered resolvin D1 (AT-RvD1), are SPMs that have been shown in the literature to possess anti-inflammatory and anti-tumorigenic attributes [28,29,30]. Previous studies have identified that dietary supplementation with fish oil or omega-3 PUFAs, as well as treatment using SPMs, effectively dampen lung inflammatory responses, and in some studies, they inhibit tumorigenesis [28,31,32,33]. We have previously reported that the use of a DHA-rich diet significantly increases RvD1 levels in the lungs of HDE-exposed mice; however, the use of this SPM or its epimer as a therapeutic strategy for alleviating agricultural organic dust-induced inflammation have yet to be examined [31]. To build upon our previous investigations and assess the carcinogenic effects of HDE, as well as provide a therapeutic strategy for combatting HDE-induced chronic inflammation, we established a chronic dust exposure mouse model of lung carcinogenesis. A/J mice were exposed to HDE for 24 weeks while receiving a weekly i.v. treatment of AT-RvD1. Upon completion of the exposure, we assessed lung tumor burden, tissue histopathology, pathway and gene-expression alterations, and screened for biomarkers related to carcinogenesis and metastasis using the A549 lung adenocarcinoma cell line. Overall, these studies demonstrate the chronic lung inflammatory and carcinogenic properties associated with agricultural organic dust exposure and highlight the use of AT-RvD1 treatment as an effective therapeutic strategy for alleviating dust-induced inflammation.

## 2. Materials and Methods

### 2.1. Chemicals & Reagents

4-(Methylnitrosamino)-1-(3-pyridyl)-1-butanone (NNK) was purchased from Sigma Aldrich (St. Louis, MO, USA). 17(R)-Resolvin D1 (AT-RvD1) was purchased from Cayman Chemicals (Ann Arbor, MI, USA). Interleukin-6 (IL-6), interleukin-10 (IL-10), and transforming growth factor beta-1 (TGFβ-1) Duoset ELISA kits were purchased from R&D Systems (Minneapolis, MN, USA).

### 2.2. Preparation of Hog Dust Extract

Agricultural organic dusts collected from swine confinement facilities were collected and prepared into aqueous extracts (HDE) as previously described [10,31]. In brief, settled dusts were collected off surfaces 1 m off the ground within indoor swine confinement facilities. These dusts were suspended in Hank’s Balanced Salt Solution at a 100 mg/mL concentration. The aqueous mixture was then centrifuged and sterile filtered at 0.22 μm to form a 100% HDE extract (vol/vol) and stored at −20 °C for later use. Using sterile phosphate-buffered saline (PBS), the 100% HDE was diluted to 12.5% HDE for all animal studies, and diluted to 1%, 2.5%, and 5% concentrations in cell culture medium for all cell-culture-related experiments. Characterization of the bacterial components of these dusts have been previously documented [3].

### 2.3. Animal Housing & Care

Male and female A/J mice 8–12 weeks of age were received from The Jackson Laboratory (Bar Harbor, ME, USA) and housed in micro-isolator cages (five per cage) at the University of California, Riverside animal barrier facilities. Mice were allowed unrestricted access to food and water, monitored for any behavioral or physiological changes, and weighed weekly (see Appendix A for weights at initial and final exposures). All experiments and procedures were approved by the University of California Riverside Institutional Animal Care and Use Committee.

### 2.4. Chronic Dust Exposure Model

This study utilized a newly optimized mouse model of chronic lung disease using long-term exposure to HDE, which was adapted from prior acute and repetitive murine dust exposure models [11,34,35]. We used a total of 20 A/J mice for the preliminary study with 10 of them being HDE-treated and the other 10 saline-treated. Each mouse was lightly anesthetized by isoflurane inhalation prior to receiving a single intranasal challenge of either 50 uL of 12.5% HDE or PBS, thrice weekly for 24 consecutive weeks. To study the tumorigenic potential of the dust in a chronic exposure setting, we used the tobacco-specific carcinogen NNK to induce lung tumorigenesis in the mice. At the start of week 4, we administered a one-time intraperitoneal injection (i.p.) of 100 mg/kg NNK, which has been well-established to generate lung adenomas in A/J mice 21 weeks post-injection [36,37]. Of the 20 exposed mice, 5 HDE-exposed and 5 saline-exposed were administered an i.p. injection of NNK, while the other 5 from each respective group received 1× PBS via i.p. injection as the control. Concluding the 24 weeks of HDE-exposure, mice were euthanized, blood was collected, and bronchoalveolar lavage fluid (BALF) retrieval was performed using three 1 mL washes with sterile saline. Each BALF wash was centrifuged, and the first aliquot was saved for cytokine analyses, while the pelleted cells from the second and third wash were combined with the first wash pellets and used to obtain total and differential immune cell counts (see Appendix A for immune cell differential example). Counts were determined microscopically on cytocentrifuge slides stained with Diff-Quik (Siemens, Newark, DE, USA). Murine cytokine levels were measured using the cell-free BALF from the first PBS wash using mouse-specific ELISA kits. We examined the levels of IL-6, IL-10, and TGFβ-1 using commercially available kits (Duoset ELISA development kits, R&D Systems). Tumors were counted and excised from the left lungs following collection and half of the left lung was then stored in RNA later (Thermo Fisher, Waltham, MA, USA) for gene expression analyses while the other half was flash frozen in liquid nitrogen and stored at −80 °C for oxylipin analysis (see Appendix A for representative images of left and right lungs with tumors). Right lungs were inflated in formalin overnight at 25 cm pressure, and visible tumors were counted the next day to achieve total lung tumor counts per mouse.

### 2.5. Lung Histopathology Scoring

Fixed lungs were embedded with paraffin and sectioned at 5 µm in preparation for subsequent staining and histological assessment. Slides were stained with hematoxylin and eosin (H&E; VWR, Radnor, PA, USA) to visualize tumor and inflammatory pathology such as lymphoid aggregate counts, bronchial and vascular inflammation, and alveolar inflammation (see Appendix A for bronchial/vascular inflammation and goblet cell hyperplasia lung tissue histology). To assess fibrotic changes within the lung tissues, we stained slides with Masson’s Trichrome (IHC World, LLC, Ellicott City, MD, USA). Goblet cell hyperplasia was determined via Alcian blue staining (IHC World, LLC., Ellicott City, MD, USA). Each of these pathological outcomes were determined by blindly scoring each slide and collecting five images, taken at objective of either 10× or 20×, of each sample, which were then assigned an individual score based on disease severity using an Echo Revolve brightfield micro-scope (San Diego, CA, USA). To assess bronchial/vascular inflammation, alveolar inflammation, and goblet cell hyperplasia, we assigned each image a number on a scale of 0 to 5 with 0 depicting no pathology and 5 representing severe pathology. Bronchial and vascular inflammation scoring criteria were determined by increased inflammation and cellularity around the lung airways and vasculature, as well as airway closure due to significant inflammation. Severe alveolar inflammation pathology was depicted by increased cellularity and reduced open area within the alveoli. Our criteria for goblet cell hyperplasia were based on how impacted the major airways were with goblet cells, with a score of 5 representing an airway fully encapsulated with goblet cells. The five images were then averaged to determine the final representative score for that respective sample. Lymphoid aggregate scoring was performed by counting the total number of lymphoid aggregates (assessed at 4× and 10×) and assigning a score of 0–5 based on the number of aggregates formed, with 0 being no aggregates and 5 being 20 or more. We defined a lymphoid aggregate as a collection of 30 or more closely packed lymphocytes. To assess changes in tissue fibrosis, we employed the Ashcroft’s score, which has been well established [38,39]. Ki-67 (1:200; #12202; Cell Signaling Technologies, MA, USA) staining to assess cell proliferation was achieved through immunohistochemistry as recommended by cell-signaling procedures (see Appendix A for representative images of lung adenomas and lymphoid aggregates expression Ki-67).

### 2.6. AT-RvD1 Treatment Study

We have previously identified that mice fed a DHA-rich diet for 4 weeks prior to receiving a single intranasal dust challenge exhibited significantly elevated levels of the SPM, resolvin D1 (RvD1), compared to dust-exposed mice administered a control diet containing no DHA [31]. To build upon our previous findings, we opted to utilize a synthetic epimer of RvD1, AT-RvD1, as a therapeutic strategy to dampen the lung inflammatory and carcinogenic outcomes identified in the HDE-exposed mice from our 24-week exposure preliminary study. AT-RvD1 carries the same anti-inflammatory and pro-resolving properties as RvD1 but has a one-log order of increased potency and is much more resistant to rapid inactivation by eicosanoid oxido-reductase (EOR), which allows it to have a longer bioactive repair process [22,40]. We utilized the same 24-week exposure model as mentioned previously; however, this time incorporated a once weekly treatment of 500 ng AT-RvD1 administered by intravenous tail injection (i.v.) beginning at the start of week 4 and ending on week 24. The AT-RvD1 was prepared into 500 ng/100 µL solution using a 100 ng/µL stock solution purchased from Cayman Chemicals (Ann Arbor, MI, USA). The control mice received 5 µL of ethanol in 95 µL of sterile saline. The dosing and frequency of administration were adapted from previous studies [29,30,41]. Other than treatment with AT-RvD1, all animal handling procedures, dosing strategies, and major outcomes were the same in this study as in the initial chronic exposure study mentioned previously, thus all data presented here will be the combination of both experimental studies. These studies were performed using male and female A/J mice which comprised 8 experimental groups: saline only, *n* = 10; HDE only, *n* = 10; saline + NNK, *n* = 9; HDE + NNK, *n* = 9; saline + AT-RvD1, *n* = 8; HDE + AT-RvD1, *n* = 6; saline + NNK + AT-RvD1, *n* = 7; and HDE + NNK + AT-RvD1, *n* = 6.

### 2.7. Oxylipin Analysis

Frozen left lung tissues were thawed and weighed, then 10 µL of antioxidant solution (0.2 mg/mL of Butylated hydroxytoluene (BHT) and Ethylenediaminetetraacetic acid (EDTA), 10 µL of internal standard mix, and 400 µL of ice-cold methanol containing 0.1% of acetic acid, were added onto the tissue samples in preparation of oxylipin analysis, as previously reported [42]. In brief, tissues were homogenized then centrifuged to retrieve the supernatant while the remaining pellets were washed with 100 µL of ice-cold methanol with 0.1% of acetic acid and 0.1% of BHT and centrifuged. The supernatants of each sample were diluted with 2 mL of H_2_O and loaded onto Oasis-HLB cartridges (Waters Corp., Milford, MA, USA) to be processed via solid-phase extraction (SPE). The cartridges (30 mg/30 μm) were washed with ethyl acetate (1 mL), methanol (2 × 1 mL), and 95:5 *v/v* water/methanol with 0.1% acetic acid (1 mL). The extracted supernatant was loaded on the cartridge and washed twice with 20% methanol. The cartridges were dried under vacuum and analytes were eluted from SPE cartridges into tubes with 250 μL of methanol followed by 1 mL of ethyl acetate into 2 mL tubes containing 6 μL of 30% glycerol in MeOH as a trap solution. The eluted fraction was dried under nitrogen and the residues reconstituted in 70 μL of methanol containing 20 nM 1-cyclohexyluriedo-3-dodecanoic acid (CUDA). Samples were vortexed for 5 min, then transferred to autosampler vials and stored at −20 °C until analysis. Following SPE, targeted oxylipin estimation was performed using a QQQ XEVO TQ-XS (Waters Corp., Milford, MA, USA) at the UC Riverside Metabolomics Core. The liquid chromatography–mass spectrometry (LC–MS) autosampler was maintained at 4 °C prior to analysis. For oxylipins analysis, injection volume of 3 μL of the extract was used. The separation was performed on the Ascentis Express column (2.1 × 150 mm, 2.7 μm particles; Sigma-Aldrich Supelco, St. Louis, MO, USA). The flow rate was maintained at 0.35 mL/min at 40 °C. Solvent A consisted of water with 0.1% acetic acid while solvent B was comprised of 90:10 *v/v* acetonitrile/isopropanol. The gradient separation method was used as follows: 26 min (0–3.5 min from 15% B to 33% B; 3.5–5.5 min B to 38%, 5–7 min to 42% B; 7–9 min to 48% B; 9–15 min to 65% B; 15–17 min to 75% B; 17–18.5 min to 85% B; 18.5–19.5 min to 95% B; from 19.5 to 21 min to 15% B, 21–26 min 15% B). The MS data was acquired on multiple reaction monitoring (MRM) mode. The electrospray ionization was performed in negative ion mode. The source and desolvation temperatures were maintained at 150 °C and 500 °C, respectively. Desolvation gas was set to 1000 L/h and cone gas to 150 L/hr and the collision gas was set to 0.15 mL/min. All gases used were nitrogen, other than the collision gas which was argon. The capillary voltage was 1 kV. Samples were normalized for relative abundance against the internal standards, while normalization for absolute quantification was performed in relation to the mg of lung tissue used from each sample. Data processing (peak integration) was performed with Skyline software (Herndon, VA, USA).

### 2.8. NanoString Gene Expression Anlyses

Of the 65 total mice used in these studies, 24 left lung tissues were randomly selected among the saline-only, HDE-only, saline + NNK, and HDE + NNK-exposed groups, for transcript and pathway gene expression analyses using the NanoString platform. Left lung tissues were homogenized and RNA was extracted using the PureLink RNA Mini Kit (Invitrogen, Carlsbad, CA, USA) and sample quality was quantified using the NanoDrop ND-100 (NanoDrop Technologies, Inc, Wilmington, DE, USA) and an Agilent 2100 Bioanalyzer (UC Riverside Core Facilities, Agilent Technologies, Santa Clara, CA, USA). We utilized the mouse PanCancer Immune Profiling Panel (NanoString Technologies, Seattle, WA, USA) codeset which is a pre-designed panel containing 770 target genes related to immune response and carcinogenesis. Sample preparation was performed by mixing 50 ng of total RNA with the codeset and reporter probes and hybridizing the mixture for 16 h to form the desired target-probe complex. The hybridized complex was then plated and run on an nCounter Sprint profiler to image and quantify the data. Following sample processing, the data were downloaded and analyzed using the nSolver software (version 4.0, NanoString Technologies, Seattle, WA, USA). Manual normalization of the expression data was performed using the geometric mean of the housekeeping genes *EEF1G*, *OAZ1*, *RPL19*, and *SDHA*. Proceeding sample normalization, 21 of the 24 samples passed the normalization parameters and were deemed viable for subsequent analyses. The samples breakdown are as follows: 6 saline-only, 4 HDE-only, 6 saline + NNK, and 5 HDE + NNK. Protein–protein interactions of differentially expressed genes were examined using the STRING database. NanoString data are deposited in NCBI’s Gene Expression Omnibus and are accessible through GEO Series accession number GSE200451 (https://www.ncbi.nlm.nih.gov/geo/query/acc.cgi?acc=GSE200451, accessed on 29 December 2021).

### 2.9. Culturing of A549 Lung Cells

Human lung adenocarcinoma-derived A549 (CCL-185) cells (ATCC, Manassas, VA USA) were grown on non-coated T-75 flasks and cultured using ATCC F-12 K medium with 10% fetal bovine serum and 1% penicillin-streptomycin in 5% CO_2_ at 37 °C. Cells were grown until at least 75% confluency, then passaged using TrypLE Express (Thermo Fisher, Waltham, MA, USA). For all HDE exposure studies, cells were seeded then treated 24 h later with either 0%, 1%, 2.5%, or 5% (vol/vol) HDE for 48 h. All preventative studies required that cells be pre-treated for 1 h using 1 ng/mL, 10 ng/mL, 20 ng/mL or 100 ng/mL of AT-RvD1 24 h after seeding, then immediately treated with 5% HDE for 48 h. Dosing strategies for all HDE and AT-RvD1 experiments were determined based on investigations from previous in vitro studies [43,44,45].

### 2.10. Western Blotting

Cell lysates were harvested from A549 cells, and proteins were resolved by SDS-PAGE and electro transferred to PVDF membranes where primary antibodies against E-cadherin (1:2000; #610181, BD Biosciences, San Jose, CA, USA), N-cadherin (1:1000; #610921, BD Biosciences, USA), and vimentin (1:500; sc-6260, Santa Cruz Biotechnology, Dallas, TX, USA) were used. Corresponding secondary horseradish peroxidase-conjugated antibodies (1:2000; Santa Cruz Biotechnology, Dallas, TX, USA) were used to detect primary antibodies on the membrane. Membranes were developed using BioRad Clarity Western ECL Substrate reagent in BioRad ChemiDoc imaging system (BioRad, Carlbad CA, USA). Protein levels were quantified by densitometry using Fiji open-source software (Image J version 1.52p; National Institutes of Health, Bethesda, MD, USA). Whole blots can be found in the Appendix A (Appendix A).

### 2.11. Bioinformatics Data Processing and Analyses

A549 cells underwent live cell imaging using a BioStation CT incubator (Nikon Instruments, Tokyo, Japan) for 48 h after being treated with HDE. We used CL-Quant (DR Vision, Seattle, WA, USA) and CellProfiler (open-source; Broad Institute, MA, USA) software, to process BioStation captured images, and MATLAB (MathWorks Natick, MA, USA) machine learning algorithm to classify cell morphology [44,46]. The time-lapse images were captured in phase contrast every 15 min for 48 h. Time-lapse images were imported into CL-Quant to segment individual cells and to process phase contrast images into binary images of cells’ silhouette. Following segmentation, binary images were imported into CellProfiler which assigned object numbers to each cell and extracted morphological features from these cells. We imported these data into MATLAB and used 11 morphological features (area, major axis length, minor axis length, maximum radius, median radius, perimeter, and numbers of nearby cells) to develop a machine learning library consisting of 300 total sampled cells. The machine learning algorithm used the training library to automatically categorize cells into cobblestone (standard A549 morphology) or elongated (mesenchymal-shaped). This library allows for a non-biased determination of cellular morphology with 97% accuracy in detection of each of the morphology classes. Total number of cells in each image was used to normalize the two categories. All data presented underwent automatic morphology classification using a combination of these softwares.

### 2.12. Statistical Analyses

All statistical analyses and graphing were performed using GraphPad Prism Software (Version 9.3.1; San Diego, CA, USA). Data are reported as the mean ± standard error of the mean for all graphs and figures. All statistical calculations performed for animal data are depicted using either two- or three-way ANOVA with Tukey’s post hoc test for multiple comparisons between the groups. Significance for all reported data was set at *p* ≤ 0.05. On all appropriate graphs, the # symbol is used to indicate significant differences in post hoc comparisons between the AT-RvD1 and non-AT-RvD1-treated groups. The * symbol is used to indicate significant post hoc differences between saline and HDE-exposed mice within each respective NNK-exposed group. All data presented underwent a Grubb’s outlier test to exclude statistical outliers. Western blot data are presented in relation to the 0% HDE-control group and t-tests using hypothetical value of 1 were used to measure significant fold change relative to normalized 0% HDE-control group. Cell morphology changes are analyzed with one-way ANOVA with Dunnett’s post hoc test. Manual sample QC and normalization for the NanoString gene expression panels was accomplished by eliminating gene probes with less than 87 counts from the data set. The threshold count was determined through background subtraction (mean +/− two standard deviations of negative controls) to screen for biological or technical variation within the sample sets without bias. All differentially expressed genes were assessed using raw *p*-values. Subsequent clustering figures were produced using nCounter Analysis and Advanced Analysis packages in nSolver 4.0 (NanoString Technologies, Seattle, WA, USA).

## 3. Results

### 3.1. AT-RvD1 Reduces Cellular Recruitment into the Airways of Dust-Exposed Mice

Our previous single and repetitive (15 instillations over 3 weeks) HDE-exposure studies have shown that HDE exposure elicits a significant increase in total and immune cell recruitment into the airways of mice [32,34]. No study has yet investigated the chronic impact of agricultural dust exposure in mice and thus, this study sought to address that gap in knowledge by examining the effects of 24 weeks of HDE exposure. Moreover, we have previously identified that both the supplemented use of a DHA-rich diet or administration of omega-3 derived SPMs to mice, dampened the dust-induced lung inflammatory response within both our single and repetitive dust exposure mouse models [31,32,34,47]. To build on our previous investigations, we opted to administer 500 ng via tail i.v. of the SPM AT-RvD1 once weekly for 21 weeks as a therapeutic strategy to mitigate lung inflammation and carcinogenesis associated with 24 weeks of HDE exposure in A/J mice (Figure 1).

We found that chronic HDE exposure, independent of NNK administration, had a significant (*p* < 0.0001) effect on total cell influx into the lung airways of mice (Figure 2A). This effect of HDE was also seen with macrophage (Figure 2B), neutrophil (Figure 2C), lymphocyte (Figure 2D), and eosinophil (Figure 2E) recruitment into the lung airways. We also saw a significant main effect of AT-RvD1 treatment within neutrophil (Figure 2C; *p* < 0.0001), lymphocyte (Figure 2D; *p* = 0.008), and eosinophil (Figure 2E; *p* = 0.0002) immune cells in the BALF. Interestingly, HDE-exposed mice treated with AT-RvD1 had significantly reduced levels of neutrophils in the BALF compared to HDE only (Figure 2C; *p* = 0.0003) and HDE + NNK-exposed mice (Figure 2C; *p* < 0.0001). Further assessment identified significant reductions in lymphocyte recruitment in HDE + NNK + AT-RvD1-exposed mice (Figure 2D; *p* = 0.007), compared to HDE + NNK-exposed mice. We also identified significant reductions in eosinophil cell influx into the lungs of HDE + AT-RvD1-exposed (Figure 2E; *p* = 0.01) and HDE + NNK + AT-RvD1-exposed mice (Figure 2E; *p* = 0.002), compared to their HDE-counterparts.

HDE exposure has been well characterized to elicit a potent pro-inflammatory response that includes the secretion of pro-inflammatory cytokines into the lung airways of exposed mice [4,48]. Alternatively, omega-3 fatty acids and their respective SPMs are known to be anti-inflammatory and thus drive an increased release of related anti-inflammatory and pro-resolving cytokines [49,50,51]. We observed a significant main effect of HDE for the secretion the pro-inflammatory cytokine IL-6 (Figure 3A; *p* = 0.01), anti-inflammatory cytokine IL-10 (Figure 3B; *p* = 0.01), and pro-EMT cytokine TGFβ-1 (Figure 3C; *p* = 0.001), in the BALF of exposed mice. We found a significant main effect of AT-RvD1 treatment and increased levels of BALF IL-10 (Figure 3C; *p* = 0.05) detected in exposed mice. Post hoc comparisons within TGFβ-1 BALF levels identified a significant increase in the HDE-exposed mice compared to the saline control mice (Figure 3B; *p* = 0.001).

### 3.2. Fatty Acid Levels in Murine Lung Tissues

Left lung tissues were harvested and used for oxylipin analysis using a pre-designed panel of fatty acids and oxylipins, such as eicosanoids and SPMs. Lung tissue samples were processed using Triple Quadrupole LC-MS following 24 weeks of exposure to HDE. Samples were normalized to relative and absolute quantification using the Skyline data processing software (Herndon, VA, USA). All oxylipins presented in Figure 4 had a significant main effect of HDE (*p* < 0.05) for relative abundance in the lung tissues, while prostaglandin D_2_ (PGD_2_) and resolvin E1 (RvE1) also had a main effect of NNK (Figure 4B; *p* = 0.02 and Figure 4H; *p* = 0.006), respectively. There was a significant main effect of AT-RvD1 for relative abundance of prostaglandin F1 alpha (PGF1a; Figure 4C; *p* = 0.009), prostaglandin F2 alpha (PGF2a; Figure 4D; *p* = 0.02), and RvE1 (Figure 4H; *p* = 0.004) in the lungs of exposed mice. Levels of PGD_2_ were significantly elevated in HDE (*p* < 0.0001) and HDE + AT-RvD1 (*p* = 0.0005) samples, compared to the saline control. We also identified a significant increase in PGD_2_ levels in HDE + AT-RvD1 exposed mice (*p* = 0.006) versus the saline + AT-RvD1-exposed group (Figure 4A). Relative abundance of prostaglandin E_2_ (PGE_2_) was significantly greater in HDE (*p* = 0.0002) and HDE + NNK (*p* = 0.03)-exposed mice, compared to their respective saline controls (Figure 4B). Post hoc comparisons did identify a significant increase in PGE_2_ levels within the HDE + AT-RvD1-exposed mice when compared to saline-exposed (*p* = 0.003) and saline + AT-RvD1-exposed (*p* = 0.02) (Figure 4B). Thromboxane B_2_ (TXB_2_) was elevated in HDE (*p* = 0.0007) and HDE + NNK (*p* = 0.008), compared to their respective saline controls (Figure 4E). We also saw an increase in TXB_2_ among HDE + AT-RvD1 (*p* = 0.01) versus saline-exposed samples (Figure 4E). The SPM, resolvin E1 (RvE1) was significantly elevated in HDE + AT-RvD1 compared to all other experimental groups (Figure 4H). NNK-exposed mice had reduced abundance of these oxylipins compared to those not administered NNK; however, this trend was not significant. Levels of PGD_2_, PGE_2_, PGF_2a_, TXB_2_, and LXA_4_ were all above the limit of detection when normalized for absolute quantification in mg of tissue.

### 3.3. Chronic Dust Exposure Enhances Lung Tumorigenesis in Mice

Male and female A/J mice aged 8–12 weeks were intranasally challenged with 50 µL of 12.5% HDE three times a week for 24 weeks. To induce lung tumorigenesis, we performed a one-time 100 mg/kg i.p. injection of the tobacco-derived carcinogen, NNK, at the start of week 4. Treatment was performed using a once-weekly 500 ng tail vein injection of AT-RvD1 beginning at the start of week 4 and extending until the end of the 24 weeks of HDE exposure. At the conclusion of the 24 weeks, mice were euthanized, and total tumor counts were assessed by combining counts from each lung. Tumors were determined to be lung adenomas through histopathological assessment. Tumor incidence and mean counts for each experimental group can be seen in Table 1.

We identified two main significant effects for tumor formation including main effects of NNK (*p* < 0.0001) and HDE (*p* = 0.001) (Figure 5). There was also a significant interaction for lung tumor formation between HDE and NNK (*p* = 0.003). Mice exposed to HDE + NNK, had significantly elevated tumors (*p* = 0.02) compared to the saline + NNK control for tumorigenesis. HDE + NNK-exposed mice treated with AT-RvD1, had significantly greater lung tumors (*p* = 0.002), compared to the baseline saline + NNK-exposed mice. HDE-exposed mice had significantly less tumors than HDE + NNK (*p* < 0.0001), saline + NNK + AT-RvD1 (*p* = 0.005), and HDE + NNK + AT-RvD1 (*p* < 0.0001)-exposed mice. There was no significance between HDE and saline + NNK-exposed mice.

### 3.4. AT-RvD1 Alters Dust-Induced Lung Histopathology Outcomes

Blinded scoring was performed to evaluate the lung tissues for histopathological outcomes including lymphoid aggregates, bronchial and vascular inflammation, goblet cell hyperplasia, alveolar inflammation, and tissue fibrosis (Figure 6). We observed a significant main effect of HDE exposure (*p* < 0.0001) for each of these outcomes of interest, and a main effect of AT-RvD1 treatment for alveolar inflammation (Figure 6D,F; *p* < 0.0001) and tissue fibrosis (Figure 6E,G; *p* = 0.0004), with each parameter exhibiting significant upregulation in HDE exposed mice vs. saline-treated mice (*p* < 0.05 for all features). Alveolar cell hyperplasia (Figure 6D,F) was significantly elevated within HDE-exposed (*p* = 0.01) and HDE + NNK-exposed (*p* < 0.0001) mice, compared to mice treated in the respective saline conditions. In HDE + NNK-exposed mice treated with AT-RvD1, alveolar hyperplasia was significantly worse (Figure 6D,F; *p* = 0.03) compared to saline + NNK-exposed mice treated with AT-RvD1. We did identify that HDE + NNK-exposed mice had significantly greater (*p* = 0.03) alveolar inflammation and cellularity compared to the HDE-only condition. HDE-exposed mice treated with AT-RvD1 did experience reduced inflammation and cellular influx in the alveolar space compared to the non-AT-RvD1-treated mice, with significance being greater in NNK-treated (*p* = 0.0005) than non NNK-treated groups (*p* = 0.01). Severe tissue fibrosis (Figure 6E,G) was observed in HDE-exposed mice compared to in those in the saline conditions within the no NNK (*p* = 0.002) and NNK-treated conditions (*p* = 0.003). Increased fibrotic changes were also identified in HDE + NNK + AT-RvD1-exposed mice compared to the saline + NNK + AT-RvD1-group (Figure 6E,G; *p* = 0.008). Ki-67 IHC staining showed expression within lymphoid aggregates of HDE-exposed mice and within lung adenomas of NNK-administered mice (Appendix A).

### 3.5. HDE Drives Increased Inflammation- and Cancer-Related Gene Expression

Transcript and pathway-level gene expression changes were assessed in the lungs of HDE- versus saline-exposed mice using the NanoString mouse PanCancer Immune Profiling Panel. We performed an advanced analysis of the 21-sample data set including six saline-only, four HDE-only, six saline + NNK, and five HDE + NNK-exposed samples, using the nSolver software. Following sample normalization, we performed a principal component analysis (Figure 7A) which showed a clear separation of HDE and saline samples, as well as clustering being driven by HDE exposure, not NNK. Furthermore, the advanced analysis identified 129 differentially expressed (*p* ≤ 0.05) genes between the HDE and the saline samples sets (Figure 7B), and three differentially expressed genes between the HDE vs. HDE + NNK-exposed mice (Figure 7C).

Using the STRING database, we input the 129 differentially expressed genes and identified significant protein–protein interactions (*p* < 1.0 × 10^−16^) for these genes. In-depth investigations into the interactions of these 129 differentially expressed genes were performed, and we identified sub-sets of these genes related to immune cell interactions, with an emphasis on macrophage functions (Table 2) and cancer progression (Table 3) among the HDE vs. saline sample sets. Differentially expressed genes for the HDE vs. HDE + NNK-exposed mice can be seen in Table 4. 

The advanced analysis generated z-scores produced from the HDE vs. saline gene expression data, which we then plotted and analyzed by two-way ANOVA to assess for significant pathway changes. Genes related to inflammation, cancer progression, innate immunity, adaptive immunity, and macrophage cell function pathways (Figure 8A–D,F), all generated a mean z-score that was associated with a significant up-regulation driven by HDE-exposure, but not NNK administration. Conversely, genes associated with cell adhesion (Figure 8E) showed a significant reduction (*p* < 0.001) in expression within the HDE-exposed group compared to the saline control group. Further examination of genes specifically related to cancer progression (Figure 9) showed increased expression changes and clustering being driven by HDE samples, independent of NNK.

### 3.6. Agricultural Dust Exposure Induces EMT in Human Lung Adenocarcinoma Cells

The results from our chronic in vivo studies suggested that dust exposure enhances lung tumorigenesis with alterations in several genes associated with epithelial to mesenchymal transition (EMT; e.g., *CHD1*, *FN1*, *TGFB1*, *VIM*). To confirm these findings, we next assessed the pro-carcinogenic EMT potential of the dust using human lung adenocarcinoma cells (A549). We used a Biostation CT Incubator (Nikon Instruments, Tokyo, Japan) to capture time-lapse images of A549 cells treated with 0% (control), 1%, 2.5%, or 5% HDE over 48 h. Time-lapse images were captured in phase contrast and imported into CL-Quant software (DR Vision, Seattle, WA, USA) to segment individual cells and convert the phase contrast image into binary images, which were imported into CellProfiler software to extract morphological features of each segmented cell. The morphological data were imported into MATLAB software (MathWorks, Natick, MA, USA) equipped with a custom machine learning algorithm to classify cells based on their morphological data [44,46]. Data processing using these bioinformatic softwares identified differences in cell morphology following treatment, including cells appearing cobblestone (standard A549 morphology) or elongated (mesenchymal-like). Cells treated with 5% HDE had significantly (*p* = 0.02) greater number of elongated cell populations, and significantly fewer number of cobblestone cells (*p* = 0.02) compared to the 0% control-group (Figure 10).

To build upon our in vivo findings identifying significant gene expression changes in *CDH1* (E-cadherin), *VIM* (vimentin), *TGFβ1*, and *TGFβR1* in dust-exposed mice compared to the control-exposed (Table 3), we assessed levels of EMT biomarkers via Western blotting in A549 cells. Furthermore, to complement our findings of altered inflammation and HDE-enhanced carcinogenesis via AT-RvD1 treatment in vivo, we assessed the impact of AT-RvD1 treatment on the EMT-associated changes identified in HDE-treated human adenocarcinoma lung cells (A549 cells). Evaluation of the carcinogenic effects of HDE was performed by treating A549 cells with 0%, 1%, 2.5%, or 5% HDE for 48 h, while assessment of AT-RvD1 was performed by pre-treating cells with 1 ng/mL, 10 ng/mL, 20 ng/mL, or 100 ng/mL of AT-RvD1 for 1 h prior to exposing cells with 5% HDE for 48 h. Western blotting was performed to evaluate protein levels of vimentin, and the tight-junction proteins E-cadherin and N-cadherin. Through these investigations, we detected a significant reduction in E-cadherin (Figure 11A,D) within the 2.5% HDE (*p =* 0.01), 5% HDE-treated group (*p* = 0.009), and 20 ng/mL AT-RvD1-treated cells (*p* = 0.004), compared to the 0% HDE control group. Expression of N-cadherin (Figure 11B,E) was significantly increased compared to the 0% control within the 5% HDE (*p* = 0.02) and 1 ng/mL AT-RvD1-treated cells (*p* = 0.04), with the 20 ng/mL and 100 ng/mL missing significance (*p* = 0.09 for both). Changes in the mesenchymal cell marker vimentin (Figure 11C,E) were seen within the 2.5% HDE (*p* = 0.03), 5% HDE-treated group (*p* = 0.04), 1 ng/mL AT-RvD1-treated group (*p* = 0.03), and 100 ng/mL AT-RvD1-treated cells (*p* = 0.01, compared to the 0% HDE control group. There were no significant differences for the 5% HDE-treated cells versus the 5% HDE + AT-RvD1-treated cells, regardless of the dose of AT-RvD1.

## 4. Discussion

The lung inflammatory and immunomodulatory effects of acute and repetitive dust exposure have been previously studied; however, the carcinogenic and chronic inflammatory damage caused by dust particulates are unclear [35,52,53]. To examine the reported anti-inflammatory and anti-tumorigenic properties of AT-RvD1, we utilized this SPM in a mouse model of chronic (24-week) dust exposure. Here, we identified a significant HDE-induced inflammation including macrophage, neutrophil, lymphocyte, and eosinophil cell influx into the lungs of HDE-exposed mice compared to the saline controls, while AT-RvD1 treatment significantly reduced immune cell recruitment. Significantly higher levels of the pro-resolving SPM RvE1 as well as anti-inflammatory cytokine IL-10 were detected in HDE + AT-RvD1-exposed mouse lungs, which could be contributing factors to the reduced inflammation identified in AT-RvD1-treated mice. These observations demonstrate the potential harm that chronic HDE exposure may cause, as well as the therapeutic benefits of AT-RvD1 treatment, which dampen inflammation caused by HDE.

Our NanoString differential expression assessment identified increased expression changes in the chemokine receptor *CXCR4* among the HDE-exposed mice. *CXCR4* is one of the most abundant chemokine receptors expressed on murine and human fibrocytes and plays a major role in the recruitment of fibrocytes to the lungs in both species [54,55,56]. This finding suggests that chronic HDE exposure is directly correlated with increased tissue damage and as such, recruitment of fibrocytes to the lungs during fibrotic disease progression. In HDE-exposed mice, we identified increased expression of *MUC1* (mucin-1), which is associated with over-production of mucin and is a hallmark of goblet cell hyperplasia, supporting the histopathology identified in the HDE-exposed lungs [57]. Treatment with AT-RvD1 significantly reduced alveolar inflammation and cellularity in both NNK and no NNK + HDE-exposed mice. The anti-inflammatory properties of AT-RvD1 were also observed in the fibrotic lung tissue of HDE-only exposed mice.

Treatment with AT-RvD1 alleviated the lung inflammatory responses induced by chronic dust exposure; however, its role in mitigating the lung tumorigenic response from chronic HDE exposure was of interest. Within our HDE are lipopolysaccharides (LPS), which comprise the Gram-negative bacteria, endotoxin, and have been argued in the literature to possess both protective or enhancement effects for lung tumorigenesis [6,7,58,59]. Epidemiological studies have identified an inverse association with reduced lung cancer risk and long-term exposure to endotoxin in work environments such as dairy, cattle, poultry, and swine farms [8,60]. Specifically, these studies found an inverse association with endotoxin exposure and risk of developing lung adenocarcinoma, which is the most prominent form of non-small cell lung cancers [60,61]. Conversely, other studies assessing population risk for lung cancer have reported that exposure to occupational organic dusts, which contain endotoxins, are associated with increased risk of lung carcinogenesis in exposed individuals [9].

Following 24 weeks of HDE exposure, we determined that the NNK-treated mice that were exposed to HDE did see significantly increased lung tumor development compared to the saline + NNK control group, indicating that HDE exposure provided an enhancement effect in a chronic setting. Interestingly, HDE + NNK-exposed mice that were treated with AT-RvD1 did not exhibit altered tumor counts compared to the HDE + NNK group. These findings differed from a recent study identifying that AT-RvD1 reduces tumor growth in the lungs of Kras^G12D^ mice, which may suggest that this change could be model specific [62]. The pro-inflammatory eicosanoids PGD_2_, PGE_2_, and TXB_2_ increase vasodilation and vascular permeability, which may not only cause increased recruitment of neutrophils and monocytes to the site of inflammation but also allow for the tumor to have the oxygen and nutrients needed to grow and survive [63,64]. PGE_2_ is an immunomodulatory agent and has been shown in the literature to suppress apoptosis and cytotoxicity of NK cells, which may also facilitate a stronger tumor microenvironment [63,65].

We propose that AT-RvD1 was ineffective at reducing HDE + NNK-enhanced tumorigenesis due to activation of a repair process promoting a M2 or tumor-associated macrophage (TAM) phenotype in the lungs of these mice. SPMs, including AT-RvD1, assist in wound repair, tissue regeneration, and promotion of the resolution of inflammation via binding to surface receptors on structural and immune cells [19,24,66]. Many of these SPMs, such as RvE1 and LXA_4_, exhibit anti-inflammatory properties that are facilitated by the clearance of neutrophils via macrophage efferocytosis, M2 macrophage polarization, and release of anti-inflammatory and tissue-repair cytokines [23,67]. Although alternatively activated M2 macrophages are considered anti-inflammatory phagocytes, many reports have suggested that these tissue-repairing immune cells can also lead to tissue over-remodeling, resulting in disease pathogenesis such as fibrosis, although we did not see enhanced fibrosis in the AT-RvD1-treated mice [68,69,70]. This can be further supported by the elevation of BALF IL-10 levels in HDE + AT-RvD1-treated mice, an anti-inflammatory cytokine that has been shown to induce M2 polarization [71,72,73]. Transcript level expression of formyl peptide receptor 2 (*FPR2)*, the main receptor in mice for RvD1, AT-RvD1, and LXA_4_, was identified in our dust-exposed mice. Expression of this receptor could provide evidence for how the HDE + AT-RvD1-treatment mediated its protective effects in dust-exposed animals [74,75]. Moreover, PGE_2_ also modulates the polarization of TAMs towards a more M2 phenotype which, would increase the secretion of growth factors and promote tumor cell proliferation [63,65]. Pathway analysis showed significant increases in expression changes related to macrophage functions in the HDE-exposed mice versus the saline controls, with increased expression changes in receptors and chemokines relating to macrophage recruitment, including *MRC1* (mannose receptor C, type 1), *CSF1R* (colony stimulating factor 1 receptor), and *CCR2* (C-C motif chemokine receptor 2). Several studies have shown that increased *CCR2* expression in tandem with elevated IL-10 expression is correlated with elevated M2 macrophage activity, further suggesting that HDE + AT-RvD1 exposure promotes M2 macrophage recruitment to the site of inflammation [70,76]. Additionally, within dust-exposed mice, we identified increased expression changes in *CHIL3* (chitinase-like protein 3; 10.9 fold increase in HDE vs. saline-treated mice) which encodes the protein Ym1, a well-known marker for M2 macrophages [77]. In one of our recent studies examining the effects of a DHA-rich diet prior to administering 3 weeks of repetitive HDE exposures in mice, we identified that HDE-exposed mice fed a DHA-rich diet displayed significantly elevated Ym1+ macrophage expression in the lungs of exposed mice, further supporting the current findings [32]. Taken together, these findings, in combination with previous literature highlighting the macrophage-polarizing effects of SPM, warrant future investigations into macrophage polarization as a mechanism underlying the protective effects of AT-RvD1 on chronic inflammation, but not lung carcinogenesis, which we observed in our model. It is plausible that tumor-dependent immunosuppression in association with the altered polarization of the immune response caused by AT-RvD1 may be responsible for the seeming disparity in these findings. However, the lack of these investigations in our current study limit our capacity to make definitive conclusions regarding this mechanism herein.

EMT is a hallmark of cancer metastasis [1,78,79]. In our assessment of gene expression changes among HDE-exposed mice, we identified markers of EMT (Table 3), including the reduced expression of the tight junction protein *CADH1* (E-cadherin), increased expression of the mesenchymal cell markers *VIM* (vimentin), *FN1* (fibronectin), growth factor *TGFB1* (transforming growth factor beta 1), and *TGFBR1* (TGFβ-1 receptor) [1,78,79,80,81]. We also identified reduced pathway expression for genes related to adhesion, which can be supported by the down regulation of *CDH1,* as seen within the HDE-exposed mice in Figure 8. We identified significantly elevated levels of TGFβ-1 in the BALF of HDE-exposed mice as well, which may provide additional evidence for an overactive repair process or serve as evidence for the promotion of EMT, as TGFβ-1 has been well-established as an inducer of EMT in various models of carcinogenesis [45,79].

To complement these observations, we further examined the carcinogenic effects of HDE exposure within human lung adenocarcinoma cells. Since the tumors formed in the NNK-administered mice were pathologically characterized as lung adenomas, we opted to use the A549 cell line to confirm the transformative effects of HDE-exposure within the tumor microenvironment. HDE-treated cells showed EMT-associated changes (Figure 10). These expression changes support the visual non-transformation to elongated or mesenchymal-like phenotypic changes identified in HDE-treated cells compared to the non-treated cells, which is a characteristic of tumor metastasis [1,78,80]. Although AT-RvD1 is better known to have a direct effect on immune cells rather than tumor cells, we utilized AT-RvD1 to identify whether pre-treatment of A549 adenocarcinoma cells with AT-RvD1 would dampen or mitigate the HDE-induced EMT response identified in these cells. Several investigations have identified that SPMs can be produced by and act on non-immune cell populations, thus providing rationale for its investigation in our study [43,44,82]. Furthermore, there are multiple documented effects of AT-RvD1 on EMT processes, including in the A549 cell line [44,82].

The in vitro data presented here accompany our findings that chronic HDE exposure enhances lung tumorigenesis by suggesting that individuals with lung cancer that endure long-term exposure to agricultural organic dust, may be at increased risk of cancer metastasis. The increase in EMT-biomarkers between both our in vivo and in vitro studies provide evidence that exposure to these dusts rather enhance lung tumorigenesis and promote a metastatic tumor environment. Our results more closely align with the literature that proposes that LPS enhances lung carcinogenesis, rather than protects against it.

## 5. Conclusions

Chronic exposure to dust from swine confinement facilities induced severe lung inflammatory and carcinogenic responses, including fibrosis, enhanced tumorigenesis, and EMT. The anti-inflammatory effects of AT-RvD1 were evidenced by reduced influx of neutrophil, lymphocyte, and eosinophils within the BALF of dust-exposed mice, increased RvE1 levels, and decreased alveolar inflammation in the lung tissues of mice. Despite the therapeutic effects of AT-RvD1 presented in these studies, treatment with this SPM did not protect against HDE + NNK-enhanced lung tumorigenesis. We propose that this response is due to an increase in M2 or tumor-associated macrophage populations in the lungs of HDE-exposed mice, which can be evidenced by the pathway and transcript alterations in prominent macrophage recruitment markers, as well as elevated IL-10 in the BALF of dust-exposed mice treated with AT-RvD1.

Our data provide a murine model that demonstrates the significant inflammatory and carcinogenic health implications that chronic agricultural dust exposure can have. The results from our studies give new insight to lung disease progression in farm workers who endure chronic exposure to agricultural organic dusts, including an increased risk for lung carcinogenesis. Furthermore, these studies propose an effective therapeutic strategy using AT-RvD1 treatment for mitigating the negative health impacts of dust exposure among individuals in the livestock and farming industries.

## Figures and Tables

**Figure 1 cancers-14-01900-f001:**
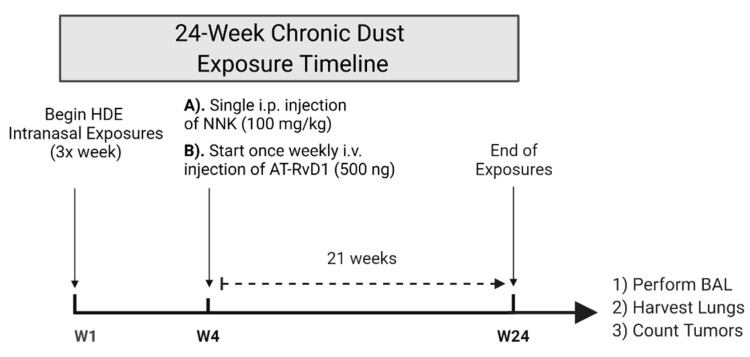
Schematic of our chronic dust exposure mouse model and dosing timeline. Male and female A/J mice received 24 weeks of intranasal challenges of 12.5% HDE, a single i.p. injection of 100 mg/kg NNK to induce tumorigenesis, and a once weekly tail i.v. of 500 ng AT-RvD1 for 21 weeks.

**Figure 2 cancers-14-01900-f002:**
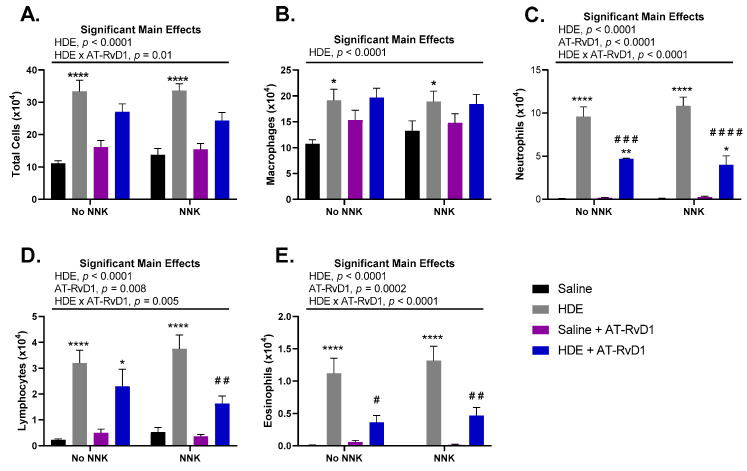
Total and immune cell counts in A/J mice following chronic HDE exposure and AT-RvD1 treatment. Following 24 weeks of HDE exposure, mice were euthanized and bronchoalveolar lavage fluid was collected and assessed for total (**A**) macrophage (**B**), neutrophil (**C**), lymphocyte (**D**), and eosinophil (**E**) cell influx into the lung airways of the mice. Main significant effects and interactions are presented above each graph as depicted from three-way ANOVA analysis. The * symbol above the HDE bars represents the statistical post hoc comparisons between the HDE- and saline-treated conditions within each respective NNK-administration groups; (* *p* < 0.05; ** *p* < 0.01; **** *p* < 0.0001). Post hoc comparisons between HDE + AT-RvD1-exposed mice and their non-AT-RvD1-treated counterparts are depicted by the # symbol above each HDE + AT-RvD1 (control and NNK) bar; (# *p* < 0.05; ## *p* < 0.01; ### *p* < 0.001; #### *p* < 0.0001).

**Figure 3 cancers-14-01900-f003:**
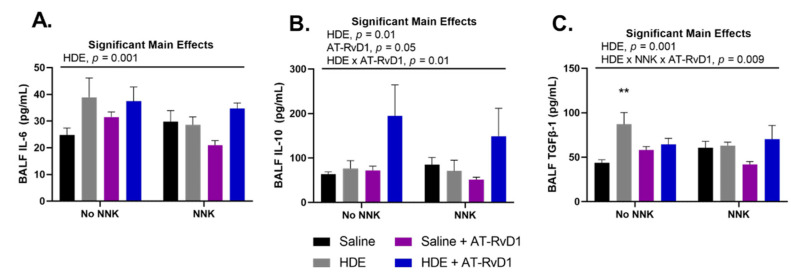
Effects of chronic HDE exposure and AT-RvD1 treatment on BALF cytokine levels after 24 weeks. Bronchoalveolar lavage was performed at the end of the 24 weeks and the fluid was collected to assess cytokine levels as described in the methods. For IL-6 (**A**), IL-10 (**B**), and TGFβ-1 (**C**); *n* = 5–10 for saline only; *n* = 7–8 for HDE only; *n* = 6–7 for saline + NNK; *n* = 6–9 for HDE + NNK; *n* = 5–8 for saline + AT-RvD1; *n* = 5 for HDE + AT-RvD1; *n* = 5–6 for saline + NNK + AT-RvD1; *n* = 4–5 for HDE + NNK + AT-RvD1. Main significant effects and interactions from three-way ANOVA analysis are shown above each graph. The ** symbol above the HDE bar represents the statistical post hoc comparison to the saline control group; (** *p* < 0.01).

**Figure 4 cancers-14-01900-f004:**
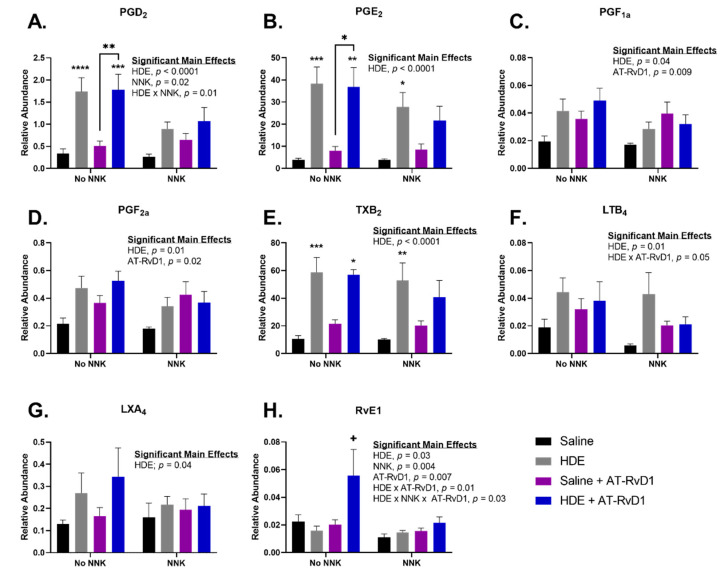
Pro-inflammatory eicosanoid and pro-resolving SPM levels in mice lung tissues following 24 weeks of HDE exposure. Oxylipin analysis was performed to assess the relative abundance of (**A**) PGD_2_ (**B**) PGE_2_ (**C**) PGF_1a_ (**D**) PGF_2a_ (**E**) TXB_2_ (**F**) LTB_4_ (**G**) LXA_4_ (**H**) RvE1 in murine lung tissue after chronic dust exposure. Three-way ANOVA with Tukey’s post hoc comparisons were performed to examine statistical changes between the experimental groups. Significant main effects and interactions of HDE, NNK, and AT-RvD1 are shown above each oxylipin. The * above the HDE samples represents significance compared to the saline-control in the respective NNK-group; (* *p* < 0.05; ** *p* < 0.01; *** *p* < 0.001; **** *p* < 0.0001). The + symbol above the HDE + AT-RvD1 bar depicts all post hoc comparisons are significant between each other experimental group; (*p* < 0.05).

**Figure 5 cancers-14-01900-f005:**
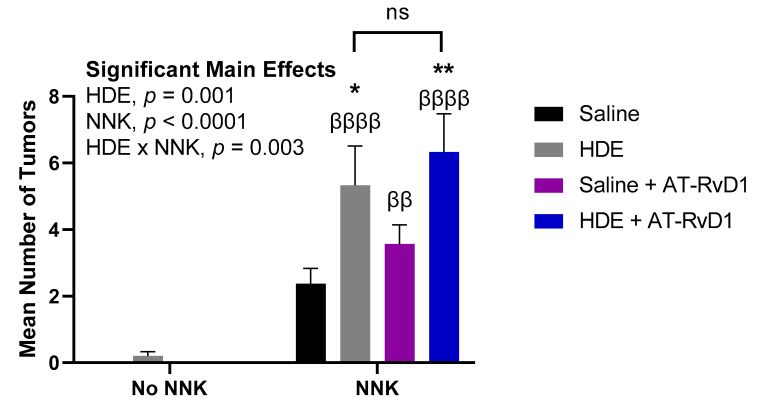
Mean lung tumor counts in mice following 24 weeks of chronic dust exposure. Right and left murine lung tissues were assessed for lung tumors following mouse euthanasia. The β symbol above each NNK sample bar represents statistical significance versus the HDE-only exposed mice (ββ *p* < 0.01; ββββ *p* < 0.0001). The * symbol above each of the bars represents statistical significance in relation to the NNK + saline-exposed control group; (* *p* < 0.05; ** *p* < 0.01).

**Figure 6 cancers-14-01900-f006:**
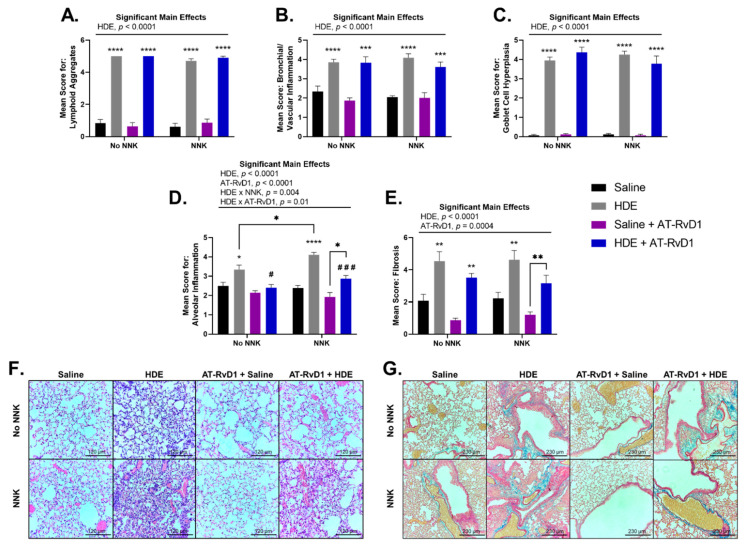
Murine lung tissue histopathology and scoring following chronic HDE exposure and AT-RvD1 treatment. Left lung tissues of mice were sectioned and blindly scored for tissue pathogenesis including (**A**) lymphoid aggregates (**B**) bronchial/vascular inflammation (**C**) goblet cell hyperplasia (**D**) alveolar inflammation (**E**) fibrosis. Lung tissues were stained with H&E to visualize (**F**) alveolar inflammation (imaged at 20×) and Masson’s Trichrome for (**G**) tissue fibrosis (imaged at 10×). Scale bars are set at 120 µm for examination of alveolar inflammation (**F**) and 230 µm for tissue fibrosis (**G**). Significant main effects and interactions of HDE, NNK, and AT-RvD1 are shown above each histological scoring parameter. The * symbol above each of the HDE bars represents statistical significance in relation to their respective saline-exposed group; (* *p* < 0.05; ** *p* < 0.01; *** *p* < 0.001; **** *p* < 0.0001). Post hoc comparisons between HDE + AT-RvD1-exposed mice and their non-AT-RvD1-treated counterparts are depicted by the # symbol above each HDE + AT-RvD1 bar; (# *p* < 0.05; ### *p* < 0.001). Direct post hoc comparisons between HDE and saline AT-RvD1-treated mice are represented by the * symbol and lines connecting those graphs.

**Figure 7 cancers-14-01900-f007:**
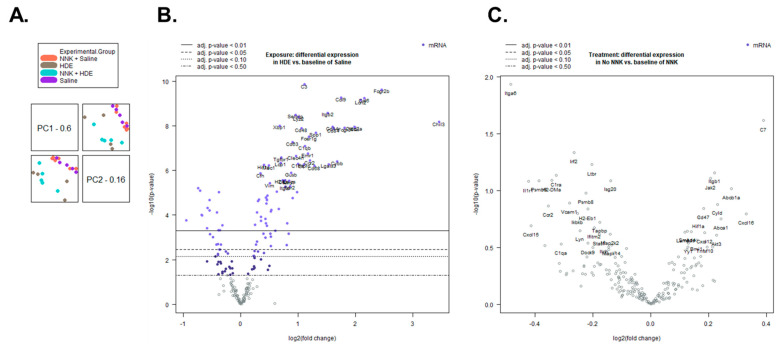
Sample clustering and differential gene expression are driven by HDE exposure. (**A**) Principal component analysis shows a separation of sample association is determined by dust, not NNK exposure. (**B**) Volcano plot of differentially expressed genes between HDE- and saline-exposed mice following 24 weeks of exposure. (**C**) Volcano plot of differentially expressed genes between HDE- and HDE + NNK-exposed mice following 24 weeks of HDE exposure.

**Figure 8 cancers-14-01900-f008:**
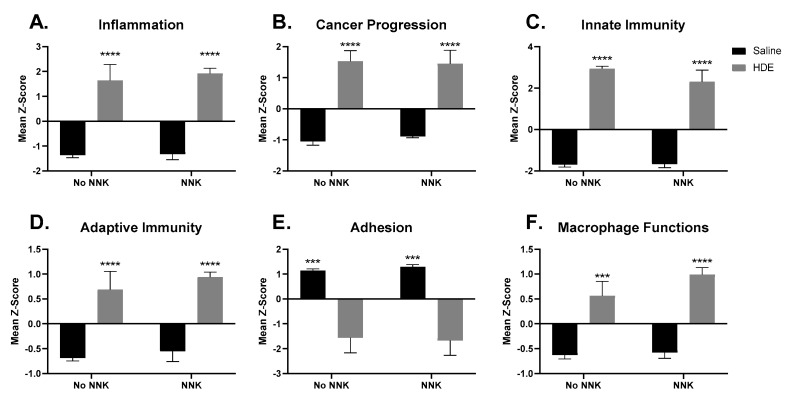
Significantly altered pathway expression changes among the 21 HDE and saline- exposed mouse lungs. NanoString Advanced Analysis was performed and the z-scores for each sample pertaining to generalized cancer and immunological-related pathways were assessed including: (**A**) inflammation (**B**) cancer progression (**C**) innate immunity (**D**) adaptive immunity (**E**) adhesion (**F**) Macrophage cell functions. The * symbol above the HDE bars represents significance against all saline-treated conditions within each respective NNK-group; (*** *p* < 0.001; **** *p* < 0.0001).

**Figure 9 cancers-14-01900-f009:**
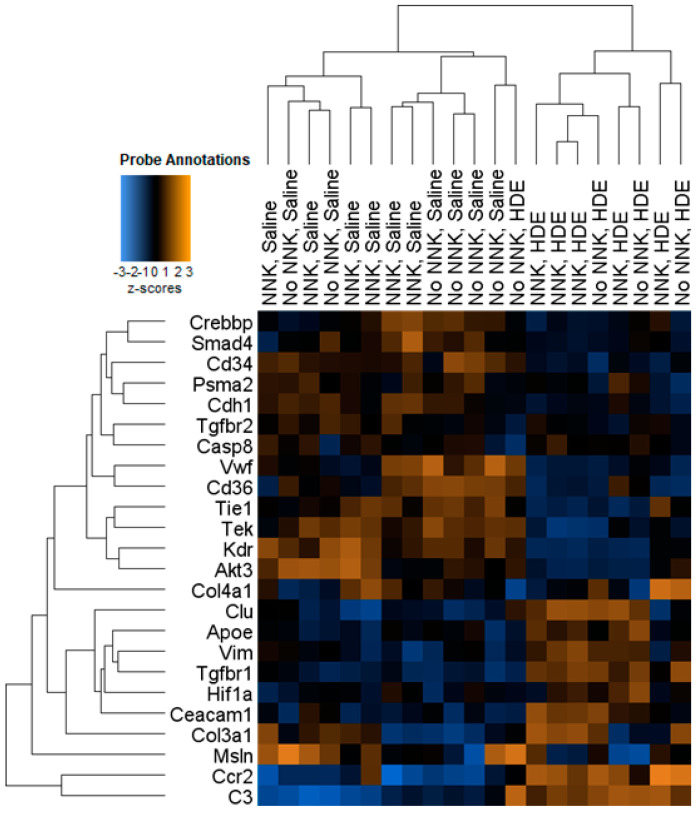
Heat map showing hierarchal clustering of cancer progression-related genes and expression changes for the 21 HDE and saline murine lung samples used in NanoString analysis.

**Figure 10 cancers-14-01900-f010:**
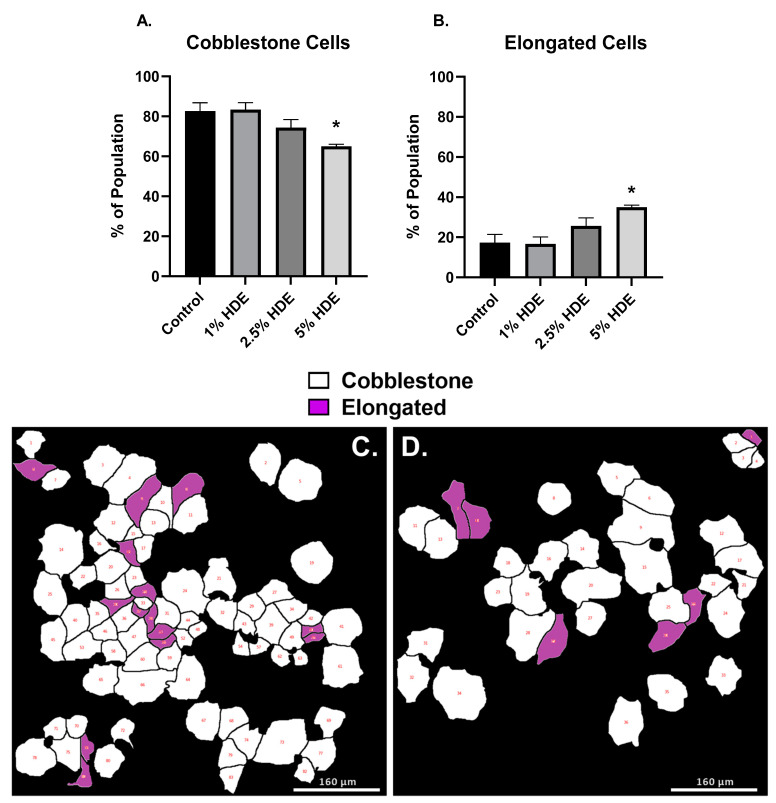
A549 cell morphological changes 48 h after treatment with HDE. One-way ANOVA was performed to assess significance for cobblestone (**A**) or elongated (**B**) cell morphology counts following 48 h of treatment with HDE. Changes in cell phenotype was determined using bioinformatic image processing software in A549 cells treated with either the 0% HDE-control group (**C**), 1% HDE (**D**), 2.5% HDE (**E**), or 5% HDE (**F**); (* *p* < 0.05).

**Figure 11 cancers-14-01900-f011:**
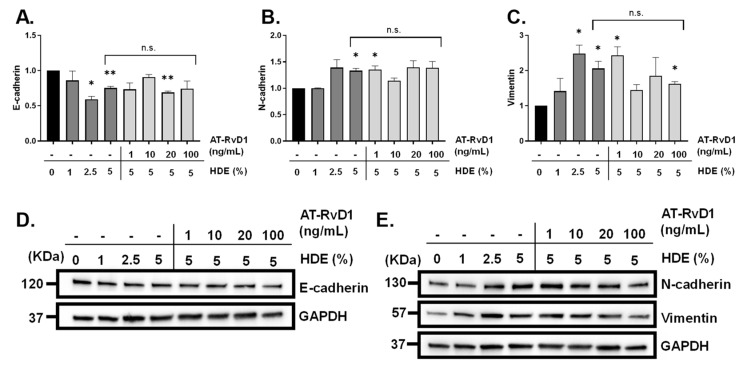
Detection of EMT markers via western blot following A549 cell treatment with HDE and AT-RvD1. Changes in protein expression of the tight junction proteins (**A**,**D**) E-cadherin (120 KDa) and (**B**,**E**) N-cadherin (130 KDa), and mesenchymal cell marker (**C**,**E**) vimentin (57 KDa). One-sample t-tests were performed relative to the 0% HDE control groups; (* *p* < 0.05; ** *p* < 0.01). Uncropped blots of Figure 11 can be found in the Appendix A.

**Table 1 cancers-14-01900-t001:** Lung Tumor Incidence and Multiplicity in A/J Mice Treated with NNK, Swine Dust Extract, and AT-RvD1.

Experimental Group	% of Mice with Lung Tumor ^a^	Lung Tumor/ Mouse ± SE ^a^	Confidence Interval (95%)
Saline Only	0	0	(0.00–0.00)
Saline + AT-RvD1	0	0	(0.00–0.00)
HDE Only	20	0.20 + 0.13	(0.00–0.50)
HDE + AT-RvD1	0	0	(0.00–0.00)
Saline + NNK	100	2.38 ± 0.46	(1.29–3.46)
Saline + NNK + AT-RvD1	100	3.57 ± 0.57	(2.17–4.97)
HDE + NNK	100	5.33 ± 1.18	(2.62–8.054)
HDE + NNK + AT-RvD1	100	6.33 ± 1.15	(3.39–9.28)

^a^ Assessed through macroscopic and microscopic examination for lung adenomas.

**Table 2 cancers-14-01900-t002:** Dust-induced Differentially Expressed Genes (*p* < 0.05) Related to Immune Cell Interactions. Arrow direction under fold change indicates up or down regulation.

Gene Symbol	Fold Change (HDE vs. Saline)
CCL6	4.43 ↑
CCL9	3.37 ↑
CCR2	2.30 ↑
CD14	3.03 ↑
CD68	2.44 ↑
CD84	3.08 ↑
CHIL3	10.9 ↑
CLEC7a	3.95 ↑
CSF1R	1.23 ↑
CXCR4	1.18 ↑
FPR2	1.81 ↑
IL13Ra1	1.35 ↑
JAK2	1.21 ↑
LCN2	4.26 ↑
MRC1	1.79 ↑
STAT1	1.25 ↓
TLR4	1.35 ↑

**Table 3 cancers-14-01900-t003:** Differential Expression of Cancer Progression Related Genes (*p* < 0.05) in HDE-exposed Mice. Arrow direction in the fold change column indicates up or down regulation.

Gene Symbol	Fold Change (HDE vs. Saline)
CD274	1.87 ↑
CD9	1.14 ↑
CDH1	1.12 ↓
FN1	1.44 ↑
KDR	1.63 ↓
LAMP1	1.24 ↑
MUC1	1.40 ↑
TGFB1	1.38 ↑
TGFBR1	1.63 ↑
VEGFA	1.68 ↓
VIM	1.42 ↑

**Table 4 cancers-14-01900-t004:** Differential Expression of Genes (*p* < 0.05) in HDE vs. HDE + NNK-exposed Mice. Arrow direction in the fold change column indicates up or down regulation.

Gene Symbol	Fold Change (HDE vs. HDE + NNK)
C7	1.40 ↓
ITGA6	1.31 ↑
IRF2	1.20 ↓

## Data Availability

Raw NanoString data are deposited at in NCBI’s Gene Expression Omnibus and are accessible through GEO Series accession number GSE200451 (https://www.ncbi.nlm.nih.gov/geo/query/acc.cgi?acc=GSE200451, accessed on 29 December 2021).

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
