# Peer review of "Aspirin-Triggered Resolvin D1 Reduces Chronic Dust-Induced Lung Pathology without Altering Susceptibility to Dust-Enhanced Carcinogenesis"

_cancers, 2022, doi:10.3390/cancers14081900_

Round 1

Reviewer 1 Report

The authors should also investigate immunosuppression. Since AT-RvD1 treatment induces immune activation in tumor-bearing mice, the authors should also investigate the effect of AT-RvD1 treatment in tumor-related immunosuppression by measuring Arg-1/IDO-1/iNOS levels. Tumor-dependent immunosuppression is vital to tumor formation and inflammation and should be investigated. 

Author Response

Reviewer 1 Comment: The authors should also investigate immunosuppression. Since AT-RvD1 treatment induces immune activation in tumor-bearing mice, the authors should also investigate the effect of AT-RvD1 treatment in tumor-related immunosuppression by measuring Arg-1/IDO-1/iNOS levels. Tumor-dependent immunosuppression is vital to tumor formation and inflammation and should be investigated. 

Author Response: Thank you for this valuable suggestion. We have now noted in the discussion section of our manuscript that this is an important limitation of our current investigation, and that this hypothesis warrants future investigations. Unforuntately, due to sample limitations and the length of time required for obtaining new samples, it is not feasible for us to complete these investigations for this manuscript in a reasonable timeframe.

Reviewer 2 Report

The authors have addressed my concerns and I do not have further comments.

Author Response

Thank you for your time, consideration, and supportive feedback.

This manuscript is a resubmission of an earlier submission. The following is a list of the peer review reports and author responses from that submission.

Round 1

Reviewer 1 Report

Journal:  Cancers

Manuscript Title: Aspirin-Triggered Resolvin D1 Modulates Lung Inflammation and Carcinogenesis In Mice Following Chronic Exposure to Organic Dust.

Manuscript Number: Cancers-1565102

Reviewer Comments:

Although this is a good research paper on lung inflammation and carcinogenesis in mice after exposure to organic dust, there are several major points that should be addressed:

1. In Results (page 8, Line 347) authors state: Chronic HDE exposure,.., had a significant effect on total cell influx into the lung airways of mice”. Immune cell influx should be studied by using immunofluorescence (IF) or Flow cytometry analysis using specified biomarkers for macrophages (CD14, CD40, CD11b), neutrophil (CD54/CD217),  lymphocytes (CD4/CD8), and eosinophils (CD11b/Integrin alpha M). BALF analysis is not adequate to support this.                                                                                                                    

2. The authors perfomed lung tumorigenesis analysis? However there are no images of the excised tumors in the mice groups, including weight, volume, survival analysis. The authors should add these data, to support their study.                                                                       

3. Since AT-RvD1 treatment induces immune activation in tumor-bearing mice, the authors should also investigate the effect of AT-RvD1 treatment in tumor-related immunosuppression by measuring Arg-1/IDO-1/iNOS levels. Tumor-dependent immunosuppression is vital to tumor formation and inflammation and should be investigated.

4. The authors claim that they perform immune analysis, however immune analysis should also include lymphocyte flow cytometry panel with actual cell counts for CD3+, CD4+, and CD8+ T cells, as well as macrophages, and B cells (which includes CD3+, CD4+, CD8+, CD19+, CD56+16+, CD45RA+, and CD45+).

5. Murine lung tissue histopathology should also include Ki-67 or CD44 marker analysis. Especially, Ki-67 is an essential marker for cell proliferation in order to determine the growth fraction of a given cell population.

Reviewer 2 Report

Current paper by Dominguez et al was set to investigate the tumorigenic effect of chronic HDE exposure in the lung and the therapeutic potential of a specialised pro-resolving mediator AT-RvD1 in treating lung inflammation and lung carcinoma. The authors demonstrated HDE initiated a robust innate immune response which aggregated the NNK-induced tumour development in mice. AT-RvD1, despite its capacity of suppressing the lung inflammation, had no effects against tumour development.

Major comments:

  1. Given the lack of effects on carcinogenesis by AT-RvD1, the title should be re-worded to more accurately reflect the findings from this study. These findings perhaps can also be discussed along with a recent paper by Vannitamby et al (Cancers, 2021), which reported a reduction of lung tumours by AT-RvD1 in a KrasG12Dmouse model.
  2. The NanoString Gene Expression Analyses should be also performed on AT-RvD1 treated NNK+HDE mice to provide a mechanistic insight on its in vivo actions and to support authors’ speculation of AT-RvD1-derived M2 macrophages/TAMs.
  3. It is unclear whether or not AT-RvD1 was able to reverse the EMT process in vitro based on current Figure 11 and the result description. What’s the justification of using AT-RvD1 in this system, since AT-RvD1 is better known to have a direct effect on immune cells rather than tumor cells. Was cell morphology analysed in the AT-RvD1 experiment?
  4. Figures 7-8 & Tables 2-3 showed a uniform effect by HDE in mice treated with or without NNK, which did not appear to explain the synergism of NNK and HDE in driving tumor development. Any measurements differentiated HDE vs HDE+NNK?

Minor comments:

  1. ‘#’ or ‘+’ symbol is recommended to be placed on AT-RvD1 groups in the graphs for clarity; Perhaps stick to ‘#’ throughout the graphs for consistency.
  2. One representative cell image is sufficient for Figure 10; Figure 10 and Figure 11 perhaps can be merged.

Reviewer 3 Report

The study under review has investigated the effects of AT-Resolvin D1 (AT-RvD1) on the modulation of lung inflammation and NNK-induced lung tumorigenesis after chronic exposure of mice to organic dust extract (dust from swine farm). It was found that chronic exposure to dust extract induces inflammation and increased numbers of lung tumors. It was concluded that AT-Resolvin D reduces inflammatory responses.

Comments:

  1. It is well known that acute and chronic exposure to organic dust in the form of aqueous dust extracts increases lung inflammation, and work by the corresponding author has shown that Maresin, a member of the specialized pro-resolving mediators (SPM), reduces lung inflammation induced by organic dust extract. Considering these the finding of induction of lung inflammation in the chronic exposure model is not new. Rationale and hypothesis for the study are not clearly stated.
  2. Many of the figures were not clear to this reviewer. Fig. 2 - I could not readily assess differences between a. HDE and HDE + AT-RvD1, b. between the same groups in the NNK treated, c. between HDE + AT-RvD1 in control and NNK mice. Have you compared these groups? May be the differences are not significant?
  3. 3 – There appears to be no effect of AT-RvD1 to reduce BALF IL-6 and TGF-beta 1 induced by HDE. Are increases in IL-6 and IL-10 significant? Did you measure KC, TNF-alpha and IL-1 beta as they are known to be important mediators of inflammation.
  4. 4 – Are the differences between various groups significant? HDE vs HDE + AT-RvD1 (control and NNK-treated groups).
  5. 5 – Is their any difference between HDE and HDE + AT-RvD1 groups (NNK)?
  6. 6 – There appears to be no differences between various groups except in the case of alveolar inflammation that is marginally increased in HDE mice treated with NNK. Could you clarify if lungs from mice used for BALF collection were subjected to histological analysis? Can you use lungs that have undergone lavage procedure for histopathological analysis? Would not lavage procedure introduce artifacts for histological analysis?
  7. RNA profiling studies – RNA expression data needs validation by qRT-PCR and western blotting for a few select genes.
  8. The choice of A549 cells as a model for alveolar type II cells is controversial.
  9. 11 – Are there differences between HDE (5%) and HDE (5%) + AT-RvD1?
  10. Discussion – the length of discussion section should be reduced, and discussion consolidated.
  11. Contrary to the title of the manuscript, it is unclear if AT-RvD1 modulates (reduces?) lung inflammation and lung tumorigenesis induced by organic dust extract.